# B-Cells and BAFF in Primary Antiphospholipid Syndrome, Targets for Therapy?

**DOI:** 10.3390/jcm12010018

**Published:** 2022-12-20

**Authors:** Lucas L. van den Hoogen, Radjesh J. Bisoendial

**Affiliations:** 1Department of Rheumatology, Radboud University Medical Center, 6525 GA Nijmegen, The Netherlands; 2Department of Rheumatology, Sint Maartenskliniek, 6525 GA Nijmegen, The Netherlands; 3Department of Rheumatology and Clinical Immunology, Maasstad Hospital, 3079 DZ Rotterdam, The Netherlands; 4Department of Immunology, Erasmus MC, 3015 GD Rotterdam, The Netherlands

**Keywords:** antiphospholipid syndrome, B-cell, plasmablast, BAFF, BLyS, rituximab, belimumab

## Abstract

Primary antiphospholipid syndrome (PAPS) is a systemic autoimmune disease characterized by thrombosis, pregnancy morbidity, and the presence of antiphospholipid antibodies (aPL). Anticoagulants form the mainstay of treatment in PAPS. A growing number of studies suggest a previously underappreciated role of the immune system in the pathophysiology of PAPS. Although B-cells are strongly implicated in the pathophysiology of other autoimmune diseases such as systemic lupus erythematosus (SLE), little is known about the role of B-cells in PAPS. Shifts in B-cell subsets including increases in plasmablasts and higher levels of BAFF are present in patients with PAPS. However, while treatment with rituximab and belimumab may ameliorate thrombotic and non-thrombotic manifestations of PAPS, these treatments do not reduce aPL serum levels, suggesting that B-cells contribute to the pathophysiology of APS beyond the production of autoantibodies.

## 1. Introduction

Antiphospholipid syndrome (APS) is a systemic autoimmune disease that is characterized by the presence of antiphospholipid antibodies (aPL) in patients with a history of thrombosis and/or pregnancy morbidity. Positivity for aPL is defined as the presence of lupus anticoagulant (LA), IgG/IgM anticardiolipin (aCL), or IgG/IgM anti-β2 glycoprotein I antibodies (aβ2GPI) above the 99th percentile of healthy controls. Thrombosis in APS may affect both arterial and venous vessels of any size. Pregnancy morbidity in APS comprises recurrent early abortions, fetal loss, and premature birth due to (pre) eclampsia or recognized features of placental insufficiency [1].

APS that is featured by LA detection and triple positivity (testing positive for LA, aCL, and anti-β2GPI) is associated with the highest risk of developing thrombosis or pregnancy morbidity [2,3,4]. APS was first identified in patients with systemic lupus erythematosus (SLE) [5,6], nevertheless not all APS patients do have SLE or another underlying autoimmune disease [7]. This has been coined as *primary* APS (PAPS) [8]. PAPS is not a forerunner of SLE or other autoimmune diseases, as only a few patients initially diagnosed with PAPS develop SLE during follow-up [9]. Approximately 1% of APS patients develop life-threatening multisite thrombosis termed catastrophic APS (CAPS), which has a mortality rate of 36% [10].

The current treatment strategy in APS, both for thrombotic and obstetric manifestations focuses on anticoagulants [11]. Vitamin K antagonists (VKA), such as warfarin and acenocoumarol are recommended to prevent recurrent thrombosis [11]. Direct oral anticoagulants (DOACs) such as rivaroxaban are not routinely used in APS, as they confer an increased risk for recurrent thrombosis, in particular arterial events [12,13]. Pregnancy morbidity in APS is managed with low-dose aspirin and low-molecular-weight heparin (LMWH) [11,14].

Aside from thrombosis and pregnancy morbidity, patients with APS may suffer from so-called ‘non-criteria’ manifestations. These include thrombocytopenia, heart valve disease, migraine, aPL nephropathy, livedo, and neurological complications like chorea, longitudinal myelitis, or seizures. These symptoms are poorly controlled by anticoagulants [15]. Despite its autoimmune nature, there is currently little evidence for the use of immunomodulatory drugs in the treatment of PAPS [16,17], with the exception of CAPS in which glucocorticoids and plasma exchange or intravenous immunoglobulins are part of the treatment regimen [11]. 

In line with anticoagulants as a cornerstone of therapy, our current understanding of the pathophysiology of APS is centered around aPL that activates endothelial cells, thrombocytes, decidual cells, and trophoblasts [18]. In the last decade, a growing number of studies have emerged that underline a previously unrecognized contribution of the activation of the immune system in the pathophysiology of APS [19]. For example, the release of neutrophil extracellular traps as well as the activation of the complement cascade and type I interferon hyperactivity are present in patients with PAPS and relate to clinical features (reviewed in [20,21,22]). 

Despite the central role of antibodies in its diagnosis and pathogenesis, relatively little is known about the role of B-cells in APS [23]. The central role of B-cells in driving autoimmunity, in general, is reflected by the success of B-cell depleting therapies such as rituximab in patients with autoantibody-mediated autoimmune diseases, most notably rheumatoid arthritis and antineutrophil cytoplasmatic antibody (ANCA) vasculitis [24]. The clinical efficacy of B-cell-depleting therapies tends to precede the decline in autoantibody levels, suggesting that the contributive role of B-cells in the pathogenesis of these diseases goes beyond the production of autoantibodies only [25]. In this review, we reconcile findings on the role of B-cells and the B-cell-related cytokine BAFF (B-cell activating factor) in the pathophysiology of PAPS and discuss the efficacy of targeting B-cells or BAFF in PAPS.

## 2. B-Cell Subsets in Primary APS

B-cells play a central role in the pathogenesis of many autoimmune and rheumatological diseases through the production of potentially harmful autoantibodies, the release of proinflammatory cytokines and other inflammatory mediators, and antigen presentation to autoreactive T-cells [25]. B-cells mature via a complex developmental program in the bone marrow which they then escape following a migration pattern in which they circulate between the peripheral blood and secondary lymphoid organs such as lymph nodes and spleen.

Upon activation of the B-cell receptor, B-cells may further mature and ultimately become plasmablasts and subsequent plasma cells, which are the major producers of (auto)antibodies. Rather than circulating in peripheral blood, plasma cells preferentially reside deep in the tissue compartments, among which are bone marrow, spleen, and connective tissues. Only those subsets of B-cells that remain present in peripheral blood have been studied in patients with PAPS

### 2.1. aPL Producing B-Cells Are Present in Healthy Subjects and Can Be Induced by Microbial Triggers

A study in three patients (without a known autoimmune disease or signs of APS) with infectious mononucleosis found that the polyclonal B-cell activation caused by Epstein-Barr virus (EBV) resulted in an increase of circulating anticardiolipin antibody expressing B-cells. A large fraction of these IgM anticardiolipin-positive B-cells expressed CD27, a marker of memory B-cells, suggesting that these cells were present prior to the EBV infection [26]. This bystander microbial activation of memory B-cells expressing aPL may explain the incident increase of aPL during an infectious disease. 

Of note, peptides derived from pathogens such as Haemophilus influenzae, Neisseria gonnorrhoeae, Streptococcus pyogenes, and cytomegalovirus share homology with or bind to β2GPI and in mice may trigger the production of aPL through molecular mimicry with β2GPI [27,28,29]. More recently the human gut commensal Roseburia intestinalis was found to have a molecular homology with β2GPI. Although the presence of Roseburia intestinalis is not unique to patients with APS, it was suggested that low levels of (gut)inflammation may result in the formation of antibodies against β2GPI through molecular mimicry with this bacterium. This hypothesis was further supported by showing that immunization of mice with Roseburia intestinalis (which is not present in mice) resulted in the production of aPL and the development of thrombosis [30].

These data suggest that bacteria or viruses may serve as triggers for the production of aPL in humans, either through molecular mimicry or through bystander activation of natural antibody-producing B-cells.

### 2.2. Total B-Cell Numbers Are Not Altered in PAPS

Whether patients with PAPS have an altered total number of B-cells has been debated in several studies. Thus, decreases [31], increases [32,33], or no changes [32,34] in the percentages of total B-cells have been reported in patients with PAPS as compared with healthy controls. Although it is unclear what causes these differences, in one study [27], the increase in percentages of B-cells was found only in patients with thrombotic APS, not in purely obstetric APS. In the other studies, patients with purely obstetric APS were not studied or at least not reported as a separate group, which hampers drawing firm conclusions on the available evidence on whether these alterations are different between thrombotic and obstetric APS.

Reporting absolute numbers of circulating cells is a more reliable method to detect changes in patients as it is not dependent on an increase or decrease in other cell types. Those studies that assessed absolute numbers of circulating B-cells in PAPS did not find differences between patients with PAPS and healthy controls [32,33,34]. Therefore the total number of circulating B-cells does not differ between patients with PAPS and healthy controls. 

In this regard, PAPS differs from SLE, in which lymphopenia (including a lower number of total B-cells) is frequently seen. Only one of these studies mentioned above also included patients with SLE, allowing a direct comparison of total B-cell numbers between SLE and PAPS. In this study, the average absolute number of circulating B-cells per microliter blood in patients with PAPS, SLE, and healthy controls was 185, 112, and 227 respectively, indeed suggesting that reduced B-cell numbers are present in SLE, not in PAPS, although not being statistically significantly different among the groups [34]. Importantly, however, a contributing factor in the decrease of B-cells in SLE may be the use of immunosuppressive drugs such as glucocorticoids [35].

### 2.3. CD5+ B-Cells Are Increased in PAPS 

The expression of CD5, an inhibitor of B-cell receptor (BCR) signaling is upregulated in self-reactive anergic B-cells and in the B-1 compartment, an innate-like B-cell subpopulation that secretes natural antibodies of relatively low affinity with potential cross-reactivity with self-antigens [36]. aPL (and other autoantibodies) have been proposed to be present in healthy individuals as natural antibodies in whom they may aid in the clearance of apoptotic cells [37]. Several studies have reported an increase in CD5+ B-cells in patients with PAPS [33,38,39,40], similar to other autoimmune diseases [41]. However, no difference in the number of CD5+ B-cells between patients with PAPS and controls has also been reported [32]. It is unknown however if the expanded CD5+ B-cell population in patients with PAPS is the source of aPL. In fact, no correlation between the number of CD5+ B-cells and aPL has been found in patients with PAPS [33].

### 2.4. Naïve B-Cells Are Increased and Memory B-Cells Are Decreased in PAPS

CD27 is a marker for memory B-cells whereas naïve B-cells traditionally express IgD, which may (unswitched) or may not (switched) be expressed during the maturation of memory B-cells. 

Three studies assessed the number of circulating naïve B-cells in patients with PAPS [32,33,34] of which two reported an increase in percentage or absolute numbers of naïve B-cells [32,33] and one showed no difference as compared to healthy controls [34]. Four studies assessed the number of peripheral memory B-cells in patients with PAPS [32,33,34,42], three of which reported decreased percentages of switched (IgD negative) memory B-cells [32,33,42] and two reported decreased percentages of unswitched (IgD expressing) memory B-cells [32,42] in patients with PAPS as compared to healthy controls. The remaining two studies reported no change in the proportions of (switched and unswitched) memory B-cells [33,34] in patients with PAPS as compared with healthy controls.

Although not consistent in all studies, patients with PAPS, show an increase in naïve and a decrease in memory B-cells. Interestingly, the expansion of the naïve compartment with a concomitant decrease in memory B-cells was confined to patients with thrombotic PAPS and not seen in patients with purely obstetric PAPS, as suggested by one study [32]. This observation indicates that an imbalance of naïve and memory B-cells may be related to thrombosis rather than the obstetric manifestations of APS, although pregnant patients were not studied. 

An increased ratio of naïve (CD27-IgD+) to memory (CD27+) B-cells has also been reported in other autoimmune diseases including Sjögren’s syndrome and systemic sclerosis [35] and is therefore not specific for PAPS. Interestingly, in one study the percentage of naïve B-cells was higher in patients with PAPS as compared with patients with SLE [33], while another study reported the opposite, namely increased numbers of naïve B-cells in patients with SLE, but not in patients with PAPS [34]. The cause of this imbalance has not been studied in PAPS but an increased differentiation of memory B-cells into plasmablasts or a disturbance in B-cell trafficking may underlie this imbalance [32].

### 2.5. Plasmablasts Are Increased in Patients with PAPS and May Be a Source of aPL in Patients with PAPS

Plasma cells are professional antibody-secreting cells that prefer to escape the peripheral blood to reside in tissue compartments and are therefore difficult to study in patients with autoimmune diseases. Plasmablasts are the forerunners of plasma cells and may be detected in peripheral blood. During their development from B-cells, plasmablasts lose the expression of CD20, the target for the B-cell-depleting drug rituximab.

Three studies measured the frequency of plasmablasts in the peripheral blood of patients with PAPS of which two reported an increase [42,43] and the other no statistically significant difference [34] in the number of plasmablasts between patients with PAPS versus healthy controls, although a similar trend was seen in this latter study. In one study, plasmablasts were identified as producers of aPL since omitting plasmablasts (by depletion of CD20 negative B-cells) from peripheral blood mononuclear cells from patients with PAPS that were stimulated ex vivo with IL-6, IL-21, CD40L, and APRIL resulted in an abrogation of the production of aPL. Nonetheless, no correlation between higher levels of aPL and higher numbers of plasmablasts was found in patients with PAPS, suggesting that other B-cells such as plasma cells are involved in the production of aPL as well [42].

Although SLE is similarly associated with an increase in plasmablasts [34,42], in SLE increased numbers of plasmablasts do correlate with anti-dsDNA antibodies and disease activity [44]. Although no clinical associations between plasmablasts numbers and clinical features were reported in patients with PAPS, a link between increased plasmablast numbers and increased activation of the type I IFN signature [42] and changes in the number of circulating regulatory and follicular helper T-cells [43] in patients with PAPS was noted.

### 2.6. Alterations of Transitional, Double Negative, and Regulatory B-Cell Subsets in PAPS

Aside from changes in the aforementioned CD5+ B-cells, naïve and (switched or unswitched) memory B-cells, and plasmablasts, other B-cell subsets have been studied in only a limited number of studies in patients with PAPS.

Transitional (CD24+CD38+) and double negative (CD27-IgD-) B-cells were found to be increased [33,43] or unaltered [34] in patients with PAPS and interleukin-10 producing regulatory B-cells (Bregs) were found decreased in patients with PAPS [42]. Transitional B-cells may develop into naïve B-cells, an increase in these cells may therefore be in line with the increase in naïve B-cells reported by others as outlined earlier. Increased numbers of double-negative B-cells are implicated in the production of autoantibodies in patients with SLE (reviewed in [45]), whether these cells are implicated in the pathogenesis of APS warrants further study. Regulatory B-cells, like their T-cell counterpart, play an immunosuppressive role through the production of IL-10 and other immunosuppressive cytokines such as TGF-β and IL-35 and are numerically and functionally impaired in SLE and other autoimmune diseases as well [46,47].

A summary of the alterations in B-cell subsets reported in patients with PAPS is presented in Table 1.

## 3. BAFF in Primary APS

B-cell activating factor (BAFF), also known as B-lymphocyte stimulator (BLyS) belongs to the tumor necrosis factor (TNF) superfamily and has been put forward as a pivotal survival and growth factor for B-cells [48,49]. The release of the 285-mer type II transmembrane protein BAFF is mediated by cleavage from the membrane by the membrane-bound protease furin. BAFF shares 30% sequence homology with another member of the TNF superfamily named a proliferation-inducing ligand (APRIL). Soluble BAFF exists as a homotrimer, but may also form heterotrimers with APRIL [50]. BAFF signals through any of its three receptors that are highly expressed by B-cells, namely BAFF receptor (BAFF-R), transmembrane activator and CAML interactor (TACI), and B-cell maturation antigen (BCMA). Except for plasma cells and centroblasts, BAFF-R is expressed by B-cells through all stages of development. TACI and BCMA may additionally bind APRIL [51]. BCMA is highly expressed in plasmablasts and plasma cells [52].

BAFF is produced by myeloid cells such as monocytes and neutrophils. Signaling through any of the aforementioned BAFF receptors results in B-cell survival and maturation, through the activation of NFκB [52]. BAFF gained an interest in the pathophysiology of autoimmune diseases when it was found that mice transgenic for BAFF (resulting in its overexpression) developed symptoms of autoimmunity, reminiscent of SLE [53]. In addition, patients with SLE were found to have elevated levels of BAFF, correlating with disease activity [52]. This ultimately led to the development of monoclonal antibodies targeting BAFF, of which belimumab is currently licensed for use in SLE [54,55].

### 3.1. BAFF Is Elevated in PAPS and Correlates with Disease Severity

Two studies compared BAFF levels between patients with PAPS and healthy controls and found higher levels of BAFF in the circulation of patients with PAPS [34,56]. Both studies also included patients with SLE and detected even higher levels of BAFF in patients with SLE. Setting the threshold at the 95th percentile of healthy controls, elevated levels of BAFF were found in 24% of patients with PAPS as compared with 40% in patients with SLE (with or without APS) [56]. From a clinical perspective, the highest levels of BAFF were found in patients with PAPS with higher adjusted global antiphospholipid antibody scores (aGAPSS) and patients with LA [56], defining patients at higher risk of thrombosis [57]. In the third study on BAFF levels in patients with PAPS, no healthy controls were included. Instead, this study included patients with SLE and rheumatoid arthritis (RA) as control groups and found no difference in BAFF levels among these groups [58].

One study specifically studied pregnant patients with obstetric PAPS and found higher levels of BAFF as compared to healthy pregnant women, particularly in those with prior adverse pregnancy events [59]. Notably, BAFF levels were lower in healthy pregnant subjects as compared with non-pregnant healthy subjects, suggesting a drop in BAFF levels during normal pregnancy [59,60]. The levels of BAFF in the pregnant obstetric PAPS patients were similar as in healthy, non-pregnant subjects [59] although pre- and post-pregnancy samples were not collected in the obstetric APS patients. In addition, this study did not assess whether higher levels of BAFF during pregnancy were associated with an increased risk of the development of obstetric complications in patients with PAPS.

### 3.2. The Expression of BAFF and Its Receptors in Monocytes and B-Cells in PAPS

Elevated mRNA expression of BAFF was found in purified monocytes of patients with PAPS, to similar levels as in patients with SLE [56]. This finding may suggest that monocytes are the source of elevated BAFF levels in PAPS, although other potential sources were not assessed. In addition, the trigger for BAFF production by monocytes in PAPS was not evaluated. The mRNA expression of the three BAFF receptors (BAFF-R, TACI, and BCMA) on total B-cells in patients with PAPS revealed no difference with healthy controls, in contrast to patients with SLE, which had higher expression of BCMA and lower expression of BAFF-R and TACI in bulk B-cells.

## 4. B-Cell Depletion in PAPS

Rituximab is a monoclonal antibody against CD20, a marker highly expressed on the majority of circulating B-cells (with the exception of plasmablasts and plasma cells). Treatment with rituximab results in the depletion of CD20-expressing B-cells which lasts for several months and is currently used for the treatment of rheumatoid arthritis, small vessel vasculitis as well as SLE [25]. Although other B-cell-depleting treatments are under development for the treatment of autoimmune diseases [24], only rituximab has been evaluated in patients with PAPS. 

Several cases report on the use of rituximab for the treatment of APS manifestations including thrombosis, thrombocytopenia, hemolytic anaemia, leg ulcers, pulmonary hemorrhage, and episodes of CAPS exist. The majority of these case reports suggest a clinical benefit of rituximab (as summarized elsewhere [61]), although this may be affected by publication bias of positive results. Only a few larger-scale studies on rituximab in PAPS exist and are described below.

### 4.1. Rituximab Does Not Reduce aPL Levels but May Ameliorate Thrombotic and Non-Thrombotic APS Manifestations

An open-label study, published in 2013, evaluated the efficacy of two doses of 1000 mg of rituximab 14 days apart in 19 patients with PAPS with non-criteria manifestations such as thrombocytopenia, cardiac valve disease, skin ulcers, nephropathy or cognitive dysfunction. Rituximab was well tolerated and positive treatment effects were noted on skin ulcers and cognitive dysfunction, whereas more variable responses were seen for the other manifestations [62]. Despite these positive results in some patient subsets, no effect of rituximab treatment on levels of aPL was seen after one year [62]. These data are in line with the previously mentioned study that observed that aPL production was related to CD20 negative plasmablasts [42] and suggests that the clinical efficacy of rituximab may be independent of the levels of autoantibodies.

With respect to thrombotic manifestations, a recent study reported no recurrent thrombotic events in 6 patients with PAPS treated with a comparable course of rituximab. However, all patients were treated with an anticoagulant as well [63]. After 18 months, none of the patients with a positive test for LA or IgG class anticardiolipin or anti-β2GPI antibodies at baseline became seronegative during follow-up, although a small reduction in the levels of these antibodies was noted in some patients [63]. As there was no control group, it is impossible to determine whether this was related to treatment with rituximab or part of the natural course of the disease. Two patients, one positive for IgM anticardiolipin antibodies and one positive for IgM anti-β2GPI antibodies, became seronegative after treatment with rituximab. However, as IgM class aPL is only weakly associated with thrombotic events [3], the importance of this finding is unknown. Analog to the previous study, aPL levels were not strongly affected by rituximab treatment in the other patients, despite clinical success.

The potentially long-lasting effect of rituximab therapy was also suggested by a case series of 6 patients with PAPS with treatment-refractory thrombocytopenia. Treatment-free remission was achieved lasting almost 4 years after a single course of rituximab treatment [64].

The largest case series on rituximab treatment in patients with PAPS describes the clinical course of 31 patients with PAPS (in addition to 9 patients with SLE and APS) in Israel [65]. Rituximab was used at various doses for the treatment of recurrent thrombosis as well as non-criteria manifestations. The majority of patients received additional immunosuppressive treatments such as glucocorticoids, plasma exchange, or cyclophosphamide. A partial or complete response to rituximab treatment, affecting cytopenias, episodes of CAPS, nephropathy, recurrent thrombosis, and skin ulcers, was reported in 80% of patients. Less pronounced effects of rituximab treatment were seen on diffuse alveolar hemorrhage and neurologic manifestations. Interestingly, all 17 patients receiving rituximab at a dose of 375mg/m^2^ once weekly for four doses (the scheme used in ANCA vasculitis) had a favorable response whereas 15/23 patients receiving rituximab at a dose of 1000 mg two weeks apart (the scheme used in RA and the former studies) had a favorable response, suggesting that the ANCA vasculitis scheme is more effective in APS. In this study, one patient treated with rituximab died of infectious complications.

In this latter study, patients with a favorable response to rituximab treatment (*n* = 13) exhibited a significant decline in IgM aCL, IgM, and IgG aβ2GPI and lupus anticoagulant levels (as determined by a decrease in Russel venom reagent ratio) although it was not reported whether this led to a negative test result in any of the patients. Contrastingly, this was not seen in patients with a partial or no response (*n* = 10). The suggestion that the clinical efficacy of rituximab is associated with a decline in aPL levels, is affected by the observation that most patients were also treated with other treatment modalities amongst which plasma exchange, and that post-treatment aPL levels were not available for all patients.

### 4.2. Rituximab Is Recommended to Use for Refractory Cases of Catastrophic APS

Rituximab is most extensively used for the treatment of CAPS. A study from 2013 on the registry of catastrophic APS patients identified 20 patients (of whom 11 with primary APS) treated with rituximab. The overall survival of the catastrophic episode was 75% in the patients treated with rituximab, slightly better than historical controls [66]. All patients were also treated with additional therapies such as glucocorticoids, plasma exchange, or intravenous immunoglobulins. Based on such observations, the recent EULAR guidelines for the management of APS have recommended the use of rituximab in refractory cases of CAPS [11].

### 4.3. CD19 CAR-T Cells in PAPS

CD19 chimeric antigen receptor (CAR) T-cells are T-cells that are engineered to target CD19-expressing B-cells and are used to treat refractory diffuse large B-cell lymphoma. CD19 CAR-T cell treatment may represent a novel way to target B-cells in autoimmune diseases and is being evaluated for the treatment of SLE [67,68]. A case report of a patient with PAPS who developed a diffuse large B-cell lymphoma was treated with CD19 CAR-T cells (after a rituximab-containing chemotherapy regimen). Treatment with CD19 CAR T-cells resulted in normalization of the IgM aCL levels, while other aPL seem to be absent [69]. Treatment CD19 CAR T-cells may represent a novel way to target B-cells in APS, however, its use will likely be limited by its high cost and severe toxicity.

### 4.4. Plasma Cell Depletion by Daratumumab in PAPS

The lack of response on aPL levels on rituximab treatment as well as the in vitro capacity of plasmablasts to produce aPL suggests that targeting plasmablasts and plasma cells is a viable option to reduce aPL levels. Daratumumab, which binds to CD38, a marker highly expressed by plasmablasts and plasma cells, results in the depletion of plasmablasts and plasma cells upon administration and is used for the treatment of multiple myeloma. To date, one case report described the use of daratumumab in addition to plasmapheresis and IVIG in a patient with PAPS with recurrent thrombotic events under adequate anticoagulant treatment. After starting daratumumab, the numbers of circulating plasmablasts, and levels of IgG aCL and IgG aβ2GPI declined, starting to reappear after approximately 4 months and no recurrent thrombotic events occurred in the following two years of follow-up [70].

## 5. Anti-BAFF Therapies in PAPS

In literature, the use of belimumab in patients with *primary* APS is limited to two reports. In the first report treatment with belimumab allowed the reduction of steroids in two patients, one with diffuse alveolar hemorrhage and one with skin ulcers as APS manifestations. In both patients, aPL titers were not affected by belimumab treatment [71]. The second report discusses an episode of catastrophic APS in a patient with primary APS in which belimumab (in addition to glucocorticoids and hydroxychloroquine) was started. Six months after initiation, IgG levels of anticardiolipin and anti-β2GPI antibodies dropped almost fourfold [72]. However, hydroxychloroquine use itself has also been associated with a reduction in aPL in patients with primary APS [73], which may thus constitute a confounding factor. An observational study on the treatment of 15 patients with PAPS with refractory symptoms with belimumab is currently underway [74]

In (NZW × BXSB)F1 mice, a mouse model for SLE and APS, treatment with BAFF-R-Ig (blocking BAFF signaling) did not prevent the development of aCL. However, treated mice had less myocardial infarction than controls, suggesting an effect of BAFF blockade on thrombosis in APS [75].

There are no controlled randomized trials on belimumab in patients with PAPS. A post-hoc analysis of the two randomized placebo-controlled trials in (a total of 1684) patients with SLE, showed no statistically significant effect of belimumab treatment on IgG or IgM anticardiolipin antibody levels [76]. However, subgroup analysis demonstrated a statistically significant drop in IgG anticardiolipin antibodies in patients taking concomitant hydroxychloroquine [76]. Of note, the majority of patients with positive IgG aCL antibodies had low to moderate levels, and other aPL were not assessed in this study. Thus far, the efficacy of other BAFF-targeted therapies such as atacicept, blisibimod, and tabalumab have not been assessed in patients with PAPS. Notably, blisibimod treatment led to a significant reduction in aCL levels in patients with SLE as compared with placebo [77].

Overall, these studies fail to show a strong effect of belimumab on aPL levels in patients with PAPS, but given its potential clinical benefits, further studies are warranted.

An overview of the clinical use of B-cell and BAFF targeting therapies in PAPS is presented in Table 2.

## 6. Conclusions

Several alterations in B-cell biology are present in patients with PAPS, including higher numbers of naïve B-cell and plasmablasts and BAFF levels as outlined above (Figure 1). The efficacy of targeted therapies by means of rituximab or belimumab has been evaluated by only a small number of uncontrolled studies, mostly involving refractory cases. Despite frequent reports of clinical benefits, aPL levels appear to be unaffected by such treatments. This suggests that the contribution of B-cells to the clinical manifestations in APS goes beyond autoantibody formation and warrants further studies on the role of B-cells in the pathogenesis of APS.

The studies summarized above have predominantly relied on the phenotypic evaluation of B-cell subsets in the peripheral blood of patients with PAPS. To our knowledge, no studies evaluated B-cellp subsets in target tissues of patients with PAPS. Only one study addressed the functional capacity of B-cells in APS and identified a decreased production of the immunoregulatory cytokine IL-10 by B-cells and that the production of aPL is limited to CD20-negative B-cells in vitro [42]. In addition, the observation of differences in B-cell subsets between patients with either thrombotic or obstetric APS as reported in one study [32] awaits confirmation in large, clinically well-defined patient groups. Further studies on the functional characteristics of B-cells, their crosstalk with other components of the pathophysiology of APS as well as the tissue distribution of B-cells are also certainly relevant and may open up avenues to new treatments.

## Figures and Tables

**Figure 1 jcm-12-00018-f001:**
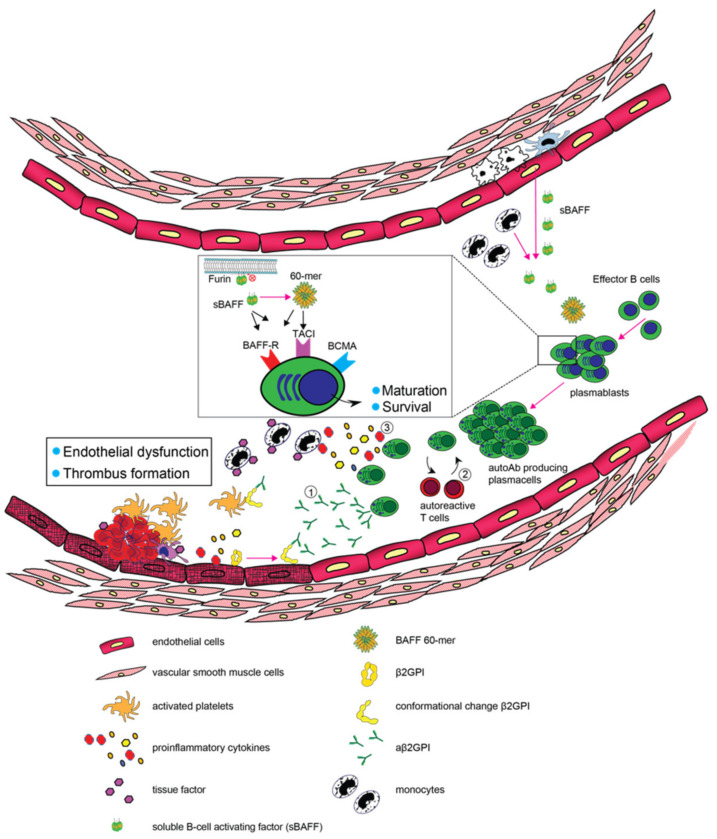
Schematic representation of the contributive role of B-cells and BAFF in the pathogenesis of APS. Growing evidence suggests that the abundance of pathogenic B-cells and BAFF may deteriorate clinical and immunological patterns in APS by promoting (1) the generation of plasma cells and production of autoantibodies, (2) stimulation of autoreactive T cells and (3) the production of pro-inflammatory cytokines.

**Table 1 jcm-12-00018-t001:** Overview of alterations in B-cell subsets reported in PAPS.

Subset	Alteration in PAPS	Reference(s)
CD5+ B-cells	Increased in PAPS	[28,33,34,35]
Naïve B-cells	Increased in PAPS	[27,28]
Unswitched Memory B-cells	Decreased in PAPS	[27,28,42]
Switched Memory B-cells	Decreased in PAPS	[27,42]
Plasmablasts	Increased in PAPS	[42,43]
Transitional B-cells	Increased in PAPS	[28,43]
Double negative B-cells	Increased in PAPS	[28,43]
Regulatory B-cells	Decreased in PAPS	[42]

As discussed in the main text, not all observations have been consistently reported among studies.

**Table 2 jcm-12-00018-t002:** Overview of clinical use of B-cell and BAFF targeting therapies in PAPS.

Drug	Clinical Use in PAPS	Reference(s)
Rituximab	Various “non-criteria” manifestations including thrombocytopenia	[61,62,64,65]
	Thrombosis recurrence	[61,63,65]
	Catastrophic APS	[61,65,66]
Daratumumab	Thrombosis recurrence (case report)	[70]
Belimumab	Diffuse alveolar hemorrhage (case report)	[71]
	Skin ulcers (case report)	[71]
	Catastrophic APS (case report)	[72]

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
