# Peer review of "B-Cells and BAFF in Primary Antiphospholipid Syndrome, Targets for Therapy?"

_jcm, 2022, doi:10.3390/jcm12010018_

Round 1

Reviewer 1 Report

In this review manuscript the authors discussed the role of B-cells and the B-cell related cytokine BAFF in primary antiphospholipid syndrome. The topic is interesting. Some concerns and suggestions are listed as below:

Any crosstalk between activate endothelial cells, thrombocytes, decidual cells, trophoblasts and B cells in antiphospholipid syndrome (APS)?

decreased [26], increased [27,28] or no changes [27,29] in the per- 88

In the section of 2.1. Total B-cell numbers are not altered in PAPS, the authors said that decreased, increased or no changes in the percentages of total B-cells have been reported in patients with PAPS as compared with healthy controls. The authors should comment why different results were noted (due to different disease stages, patients' age, power, therapeutic drugs or differnet measurements)? How about changes of T cell numbers in these patients?

Why total B-cell numbers are not altered in PAPS?

In the section of 2.2. CD5+ B-cells are increased in PAPS and may produce aPL as natural antibodies, the authors concluded that it is unknown if the expanded CD5+ B-cell population in patients with PAPS is the source of aPL?

The section of 2.3. aPL producing B-cells are present in healthy subjects seems unnecessary?

In line 152, what do you mean by saying 'patients with PAPS therefore tend to show'?

Regarding IL-10-Bregs, different subsets of Bregs and Bregs‐associated molecules TGF‐β, and IL‐35 should not be ignored (J Neurosci Res. 2016 Aug; 94(8): 693–701).

The role of B-cells and the B-cell related cytokine BAFF in primary antiphospholipid syndrome can be summarized using a figure.

This finding may suggest that monocytes are the source of elevated BAFF levels in PAPS. How about changes of circulating monocytes in PAPS?

How will BAFF expression be changed following B-cell depletion by Rituximab in PAPS and other diseases?

I wonder if the function of B cells has been changed in the condition of PAPS. This is a major concern.

Future research directions should be discussed.

Author Response

Dear Editor,

We thank the reviewers for their thorough and critical assessment of our manuscript. We have adjusted the manuscript based on their suggestions and feel it has substantially improved our manuscript. We hope you will now find it suitable for publication in Journal of Clinical Medicine.

Please find below our point-by-point reply to the concerns raised by the reviewers.

Reviewer 1

Comment: Any crosstalk between activate endothelial cells, thrombocytes, decidual cells, trophoblasts and B cells in antiphospholipid syndrome (APS)?

Reply: To our knowledge, no studies have directly studied whether a cross-talk between B-cells and other components of the pathophysiology of APS exist. The topic has been touched upon lightly by two studies cited in our manuscript, as reported in section 2.6. In the first study, an association between increased plasmablast numbers and a SNP associated with type I IFN signaling was found, suggesting a cross-talk between plasmablasts and type I interferon. In the second study, increases in plasmablasts were associated with alterations in regulatory and follicular helper T-cells in patients with APS. However, whether these changes are a direct consequence of cross-talk between these cells has not been studied in PAPS. We agree with the reviewer that this is a scientific lacuna and have incorporated this in our manuscript as a suggestion for further research.

Comment: In the section of 2.1. Total B-cell numbers are not altered in PAPS, the authors said that decreased, increased or no changes in the percentages of total B-cells have been reported in patients with PAPS as compared with healthy controls. The authors should comment why different results were noted (due to different disease stages, patients' age, power, therapeutic drugs or differnet measurements)? How about changes of T cell numbers in these patients?

Reply: We agree with the reviewer that it is interesting to study what causes the difference in % of B-cells between studies. However, based on the available data firm conclusions on this matter cannot be made. In the study by Carbone et al., the increase in the percentage of B-cells was only found in patients with combined thrombotic and obstetric APS, not in patients with purely obstetric APS, suggesting this is related to thrombotic APS, although all other studies also included patients with thrombotic APS and did not report such a change. We have adjusted the manuscript in section 2.1.

Although beyond the topic of our review, percentage of total CD3+ T-cells in patients with PAPS were found unchanged in the study by Dal Ben et al and Carbone et al., not reported in the study by Alvarez-Rodriguez et al. and decreased in the study by Simonin et al. As the study by Simonin et al is among the studies reporting an increase in % of B-cells, this indeed suggests that their finding is influenced by changes in other cell types.

Comment: Why total B-cell numbers are not altered in PAPS?

Reply: The absence of a decrease in B-cells in PAPS is in contrast to SLE in which lower number of B-cells are frequently reported. Although it is unknown what causes this difference we speculate that this is due to differences in their pathophysiology, with autoimmune cytopenias in general being a prominent feature of SLE but not of PAPS. In addition, as mentioned in our manuscript, immunosuppressive drugs including glucocorticoids which are not typically recommended in PAPS may cause a decrease in B-cells in SLE.

Comment: In the section of 2.2. CD5+ B-cells are increased in PAPS and may produce aPL as natural antibodies, the authors concluded that it is unknown if the expanded CD5+ B-cell population in patients with PAPS is the source of aPL?

Reply: We agree with the reviewer that the subheading is inconsistent with its content and have adjusted the subheading accordingly by removing the latter part.

Comment: The section of 2.3. aPL producing B-cells are present in healthy subjects seems unnecessary?

Reply: Section 2.3 discusses the presence of aPL positive B-cells in healthy individuals and how they may be triggered. As reviewer number 3 made a similar comment we have adjusted the placement of this section.

Comment: In line 152, what do you mean by saying 'patients with PAPS therefore tend to show'?

Reply: We aimed to state that, although not statistically significant among all studies, overall the studies point toward an increase in naïve B-cells. As the reviewer points out that this sentence is unclear we have decided to remove “tend to” to prevent misinterpretation.

Comment: Regarding IL-10-Bregs, different subsets of Bregs and Bregs‐associated molecules TGF‐β, and IL‐35 should not be ignored (J Neurosci Res. 2016 Aug; 94(8): 693–701).

Reply: We thank the reviewer for his suggestion regarding the presence of Breg subsets and their associated cytokines although such subsets of Bregs have not been studied in the context of APS. We have mentioned the existence of these subsets and their relevance to autoimmune diseases referring to the suggested reference by the reviewer in section 2.7.

Comment: The role of B-cells and the B-cell related cytokine BAFF in primary antiphospholipid syndrome can be summarized using a figure.

Reply: Following the suggestion of the reviewer we have added a figure regarding the role of B-cells and BAFF in PAPS to the revised manuscript.

Comment: This finding may suggest that monocytes are the source of elevated BAFF levels in PAPS. How about changes of circulating monocytes in PAPS?

Reply: Although beyond the scope of our manuscript, shifts in monocyte subsets have been reported in patients with PAPS and other autoimmune diseases. The study on monocyte mRNA BAFF expression was performed by the first author of the current manuscript; in the same sample we did assess the relative frequency of monocyte subsets based on the expression of CD14 and CD16 (classical, intermediate and non-classical monocytes). Although not reported in our original manuscript, we did not observe a correlation between higher levels of BAFF monocyte mRNA and changes in monocyte subsets.

Comment: How will BAFF expression be changed following B-cell depletion by Rituximab in PAPS and other diseases?

Reply: To our knowledge, in patients with PAPS the kinetics of BAFF levels after rituximab treatment have not been assessed. However, in other autoimmune disease it has been shown that rituximab leads to a surge in BAFF levels (for instance in Sjogren’s: Pollard R, Ann Rheum Dis 2013); this is also the rationale behind combining rituximab and belimumab treatment in autoimmune diseases. A phase 2 study in patients with lupus was published in 2021 showing positive results (Shipa Ann Intern Med 2021) and a phase 3 clinical trial on this approach is currently enrolling patients (van Schaik Trial 2022). We expect that if the results of these studies turn out to be successful this approach will also be attempted in patients with PAPS.

Comment: I wonder if the function of B cells has been changed in the condition of PAPS. This is a major concern.

Reply: We agree with the reviewer that this is a major lack in our understanding of the role of B-cells in PAPS. Only one study assessed the production of cytokines, in this case IL-10, after stimulation of B-cells of PAPS to report alterations in regulatory B-cells in APS. The same study also identified plasmablasts as producers of aPL in vitro. In line with the following suggestion by the reviewer we have listed this as a future research direction in APS.

Comment: Future research directions should be discussed.

Reply: We have expanded our discussion as suggested by the reviewer

Reviewer 2 Report

The authors summarized the association between B-cells and antiphospholipid syndrome, and the effectiveness of immunosuppression like rituximab. The manuscript was written well. I just wrote some minor comments.

The association between B-cells and antiphospholipid syndrome is written well. The previous studies are well summarized. However, there are too much information. So, please summarize the information in a table for readers' understanding.

Although the authors described the effectiveness of immunosuppression in the sentences, it would be also better to summarize the information and the characteristics of drugs in a table. to compare some drugs.

Author Response

Dear Editor,

We thank the reviewers for their thorough and critical assessment of our manuscript. We have adjusted the manuscript based on their suggestions and feel it has substantially improved our manuscript. We hope you will now find it suitable for publication in Journal of Clinical Medicine.

Please find below our point-by-point reply to the concerns raised by the reviewers.

Reviewer 2

The authors summarized the association between B-cells and antiphospholipid syndrome, and the effectiveness of immunosuppression like rituximab. The manuscript was written well. I just wrote some minor comments.

Comment: The association between B-cells and antiphospholipid syndrome is written well. The previous studies are well summarized. However, there are too much information. So, please summarize the information in a table for readers' understanding.

Reply: We thank the reviewer for his suggestion to summarize the information from paragraph 2 in a table. We have added Table 1 as shown below in our manuscript.

Comment: Although the authors described the effectiveness of immunosuppression in the sentences, it would be also better to summarize the information and the characteristics of drugs in a table. to compare some drugs.

Reply: following the suggestion of the reviewer we have summarized the findings regarding the changes in B-cell subsets in a table and added it to our revised manuscript

Reviewer 3 Report

Reviewer opinion

This review comprehensively described B cell and BAFF alterations, and clinical efficacy of B-cell and BAFF-targeting therapies in PAPS patients. The explicit summary is useful for physicians who treat APS. There are some minor points to be addressed.

1.      As the authors mentioned in the text, obstetric and thrombotic APS may have different B cell alterations due to disparate underlying mechanisms. Please specify the study population (obstetric or thrombotic APS), if applicable, when describing study results in APS patients.

2.      Page 3 Line 98, please cite the reference. In addition, there was no significant difference in the circulating B cells among the different groups (p = 0.184). Please do not over-interpret the study result.

3.      Section 2.3 should be moved to before section 2.1 since it deals with a somewhat different issue when compared with other sections (alterations in B cell subsets of PAPS patients).

4.      Page 4 Line 150, the absolute number and percentage of switched, rather than unswitched, memory B cells are significantly decreased in pAPS VTE patients relative to controls (reference 28). Please verify.

5.      Follicular helper and follicular regulatory T cell are crucial in the generation of humoral immunity. Could the authors summarize the literature with regards to alterations of these T cells in PAPS patients?

6.      The heading of Section 4 should be revised to B cell-depleting therapies in PAPS, since not only rituximab is mentioned in the section.

7.      The heading of Section 5 should be revised to anti-BAFF therapies in PAPS, since not only belimumab is mentioned in the section.

8.      Page 8 Line 355, please cite the reference.

9.      In Section 2.4, please give some explanations about why naïve B-cells are increased and memory B-cells are decreased in PAPS patients.

10.  The sample sizes of some studies regarding B cells in PAPS patients are relatively small. This limitation should be mentioned in the Conclusion.

11.  Some typos should be corrected: Line 134, “R”oseburia, Line 162, “SLE patients”

Author Response

Dear Editor,

We thank the reviewers for their thorough and critical assessment of our manuscript. We have adjusted the manuscript based on their suggestions and feel it has substantially improved our manuscript. We hope you will now find it suitable for publication in Journal of Clinical Medicine.

Please find below our point-by-point reply to the concerns raised by the reviewers.

Reviewer 3

This review comprehensively described B cell and BAFF alterations, and clinical efficacy of B-cell and BAFF-targeting therapies in PAPS patients. The explicit summary is useful for physicians who treat APS. There are some minor points to be addressed.

  1. As the authors mentioned in the text, obstetric and thrombotic APS may have different B cell alterations due to disparate underlying mechanisms. Please specify the study population (obstetric or thrombotic APS), if applicable, when describing study results in APS patients.

Reply: Unfortunately this was only assessed in one study, with the others either not studying purely obstetric APS patients or not reporting the results for them separately. We agree with the reviewer that this is relevant and have adjusted the manuscript to mention that this has not been assessed by others and have listed it as a suggestion for further research in the concluding remarks.

  1. Page 3 Line 98, please cite the reference. In addition, there was no significant difference in the circulating B cells among the different groups (p = 0.184). Please do not over-interpret the study result.

Reply: We have added the correct reference. The p-value that the reviewer refers to is the overall P-value of the Kruskall-Wallis test of the comparison of healthy controls with PAPS and SLE; the (uncorrected) p-values of the separate comparisons were not reported to our knowledge. We have added the fact that the comparison was not statistically significant.

  1. Section 2.3 should be moved to before section 2.1 since it deals with a somewhat different issue when compared with other sections (alterations in B cell subsets of PAPS patients).

Reply: As a similar suggestion was made by reviewer 1 we have moved this section to a separate paragraph in 2.1 as suggested by the reviewer (and changed the numbering of the other sections accordingly).

  1. Page 4 Line 150, the absolute number and percentage of switched, rather than unswitched, memory B cells are significantly decreased in pAPS VTE patients relative to controls (reference 28). Please verify.

Reply: As reported by the reviewer, a few lines earlier indeed we report the observation of decreased switched memory B-cells in the study by Simonin et al. Although the study reports a similar trend in unswitched memory B-cells we have now adjusted the manuscript to state this correctly, removing that a similar trend is found.

  1. Follicular helper and follicular regulatory T cell are crucial in the generation of humoral immunity. Could the authors summarize the literature with regards to alterations of these T cells in PAPS patients?

Reply: An overview of studies on follicular helper T-cells in PAPS is beyond the scope of our review. In addition to the study cited in our manuscript a recent study by Zang et al (Clin Exp Med 2022) reports changes in follicular helper and follicular regulatory T-cell subsets in APS. However, B-cells were not assessed in this study and therefore not included in our manuscript.

  1. The heading of Section 4 should be revised to B cell-depleting therapies in PAPS, since not only rituximab is mentioned in the section.

Reply: we have adjusted the section heading according to the reviewer’s suggestion

  1. The heading of Section 5 should be revised to anti-BAFF therapies in PAPS, since not only belimumab is mentioned in the section.

Reply: we have adjusted the section heading according to the reviewer’s suggestion

  1. Page 8 Line 355, please cite the reference.

Reply: We apologize, the line numbers in the version for reviewers does not seem to correspond with our version. We will ask the editorial team to clarify where the reference should be added.

  1. In Section 2.4, please give some explanations about why naïve B-cells are increased and memory B-cells are decreased in PAPS patients.

Reply: although not formerly studied, alterations in B-cell trafficking or plasmablast differentiation may underlie the imbalance in naïve and memory B-cells, as suggested by the study by Carbone et al. We have added this as a potential explanation in this section, referring to the study by Carbone et al.

  1. The sample sizes of some studies regarding B cells in PAPS patients are relatively small. This limitation should be mentioned in the Conclusion.

Reply: we have adjusted the conclusion to mention that studies in larger cohorts are necessary for future studies

  1. Some typos should be corrected: Line 134, “R”oseburia, Line 162, “SLE patients”

Reply: we have adjusted the manuscript accordingly.

Round 2

Reviewer 1 Report

The authors have addressed my concerns.